# m^6^A Demethylase ALKBH5 Restrains PEDV Infection by Regulating *GAS6* Expression in Porcine Alveolar Macrophages

**DOI:** 10.3390/ijms23116191

**Published:** 2022-05-31

**Authors:** Jian Jin, Chao Xu, Sen Wu, Zhengchang Wu, Shenglong Wu, Mingan Sun, Wenbin Bao

**Affiliations:** 1College of Animal Science and Technology, Yangzhou University, Yangzhou 225009, China; dz120210008@yzu.edu.cn (J.J.); mx120200807@stu.yzu.edu.cn (C.X.); senwu1993@163.com (S.W.); zcwu@yzu.edu.cn (Z.W.); slwu@yzu.edu.cn (S.W.); 2Institute of Comparative Medicine, College of Veterinary Medicine, Yangzhou University, Yangzhou 225009, China

**Keywords:** ALKBH5, PEDV, alveolar macrophages, N^6^-methyladenosine, *GAS6*, pig

## Abstract

Porcine epidemic diarrhea virus (PEDV) is a burdensome coronavirus for the global pig industry. Although its fecal-oral route has been well-recognized, increasing evidence suggests that PEDV can also spread through airborne routes, indicating that the infection may also occur in the respiratory tract. N^6^-methyladenosine (m^6^A) has been known to regulate viral replication and host immunity, yet its regulatory role and molecular mechanism regarding PEDV infection outside the gastrointestinal tract remain unexplored. In this study, we demonstrate that PEDV can infect porcine lung tissue and the 3D4/21 alveolar macrophage cell line, and the key m^6^A demethylase ALKBH5 is remarkably induced after PEDV infection. Interestingly, the disruption of ALKBH5 expression remarkably increases the infection’s capacity for PEDV. Transcriptome profiling identified dozens of putative targets of ALKBH5, including *GAS6*, which is known to regulate virus infectivity. Further, MeRIP-qPCR and mRNA stability analyses suggest that ALKBH5 regulates the expression of *GAS6* via an m^6^A-YTHDF2-dependent mechanism. Overall, our study demonstrates that PEDV can infect porcine lung tissue and 3D4/21 cells and reveals the crucial role of ALKBH5 in restraining PEDV infections, at least partly, by influencing *GAS6* through an m^6^A-YTHDF2-dependent mechanism.

## 1. Introduction

Porcine epidemic diarrhea (PED) is caused by the porcine epidemic diarrhea virus (PEDV), and infected piglets are characterized by severe atrophic enteritis, watery diarrhea, dehydration, vomiting, and high mortality [1]. In Asia and Europe, as the main pork production regions in the world, the large-scale outbreak and spread of PEDV have caused huge economic losses to the pork industry [2]. In recent years, PEDV has exhibited a continuously spreading trend between pig farms and shown a higher transmission potential, compared to other seasonal gastrointestinal viruses [3]. Most contemporary research on the pathogenesis of PEDV is limited to the gastrointestinal tract. However, a few studies have identified an airborne route of PEDV [4], indicating that the infection may also occur in the respiratory tract and lungs. In addition, the variations of PEDV, immunity avoidance, traditional prevention and control strategies, and vaccines are insufficient for providing sustainable and effective protection for piglets [5,6,7]. As an intracellular parasitic virus, complex interference measures have been formed to evade the defense mechanism of host cells, and increasing studies have confirmed that viruses can destroy their normal immunity function by affecting the epigenetic modification process of the host cells.

RNA modifications, which play important roles in the regulation of many biological processes, have received increasing attention in recent years [8]. Among them, N6-methyladenine (m^6^A) is the most abundant form [9], and it exists not only in the transcripts of eukaryotic and DNA viruses but also in the genomes of RNA viruses [10]. m^6^A modifications are highly reversible, with their patterns established by so-called “writers”, including the METTL family proteins and Wilms’ tumor 1-associating protein (WTAP) [11], and removed by “erasers”, including the obesity-associated protein (FTO) and alkB homologue 5 (ALKHB5) [12,13]. Furthermore, the YTH domain family (YTHDF) protein serves as the “reader”, which selectively binds methylated RNA molecules and mediates downstream processes [14,15]. m^6^A modifications play important roles in viral replication and host immunity responses. For example, silencing ALKHB5 in human T cells increases the replication of human immunodeficiency virus type 1 (HIV-1) [16], and the inactivation of METTL3 in A549 cells inhibits influenza A virus (IAV) replication [17]. Other studies demonstrate that the overexpression of YTHDF2 promotes the replication of simian virus 40 (SV40) [18], while m^6^A modifications inhibit the lytic replication of ZIKV [19]. Recent evidence indicated that m^6^A modifications modulate the replication of PEDV [20], yet that study was based on results from green monkey Vero kidney cells and porcine LLC-PK1 kidney epithelial cells. However, the mechanisms of m^6^A modification for PEDV infection in other tissues or cell types outside of the gastrointestinal tract remain to be explored.

In this study, we investigated the function of m^6^A and its associated modifying enzymes during PEDV infection, using porcine lung tissue and alveolar macrophages (3D4/21) as a model. We first analyzed the differential expression of different m^6^A modifiers in piglet lung tissues after PEDV infection using qPCR; then, we analyzed the effect of ALKBH5 demethylase expression levels on PEDV replication using RNA interference, Western blot, and copy number determination at the cellular level. Finally, we screened the candidate genes regulated by ALKBH5 targeting using transcriptome sequencing and validated the effect of ALKBH5-mediated m^6^A on target gene expression levels using qPCR, MeRIP-qPCR, and Western blot assays. Collectively, these results revealed the relationship between the m^6^A demethylase ALKBH5 and PEDV resistance and its regulatory mechanism for PEDV infection.

## 2. Results

### 2.1. PEDV Infection Promotes the Expression of ALKBH5 in the Lung Tissues of Piglets

Inspired by recent findings that PEDVs also spread through airborne routes [4], we first examined whether PEDV infection occurred in lung tissue. Lung tissue samples, which were collected from five piglets with typical PEDV infection symptoms and five normal piglets, were analyzed by RT-PCR against different pathogens, including PDCoV, PoRV, TGEV, and PEDV. The results confirmed the existence of PEDVs, but not other viruses, in the lung tissue samples of five diarrheal piglets (Figure 1a), suggesting that PEDV could infect the lung tissues. We further examined how PEDV infection could affect the expression of different m^6^A modifiers, including the methyltransferases METTL3, METTL14, and *WTAP* and demethylases *FTO* and *ALKBH5*. The results showed that only *ALKBH5* was significantly induced by PEDV infection (*p* < 0.001) (Figure 1b). The expression of *ALKBH5* was also compared in twelve different tissues of 7-day-old healthy piglets, which showed the highest expression in the lungs, followed by the spleen, jejunum, duodenum, kidney, and ileum (Figure 1c). Together, our results confirm that PEDV could, indeed, infect the lungs; our results also demonstrate that the m^6^A demethylase ALKBH5 has the highest abundance in lung tissues, which can be further induced after PEDV infection.

### 2.2. PEDV Infection Up-Regulates ALKBH5 Expression in 3D4/21 Alveolar Macrophage Cell Line

After revealing the induced expression of *ALKBH5* by PEDV infection in the lungs, we further examined PEDV infection in the 3D4/21 alveolar macrophage cell line. Normal 3D4/21 cells were fusiform, with clear outlines and obvious boundaries, and they were uniformly distributed on the cell plate. After being infected with a classical PEDV strain, CV777, for 36 h, they exhibited typical cytopathic effects, which were represented by the significant shrinkage, rounding, and loss of normal morphology (Figure 2a). RT-PCR confirmed that the CV777 strain could infect 3D4/21 cells (Figure 2b). qPCR was used to detect the expression of apoptosis and inflammation-related genes, which showed that *Caspases 3/8/9* were significantly up-regulated after infection for 36 or 48 h, and the *Bcl-2/Bax* ratio was significantly decreased after infection for 36 h (Figure 2c). The expression levels of inflammation-related genes, including *IL-6*, *IL-12*, *IFN-β*, and *IL-1β*, were also significantly increased after infection for 36 or 48 h (Figure 2d). Similar to the lung tissues, the expression levels of *ALKBH5* were significantly up-regulated after infection for 36 h and 48 h (Figure 2e). Western blot further confirmed that the protein abundance was also remarkably increased (Figure 2f). Together, these results suggest that, similar to that observed in lung tissues, PEDV can also infect the 3D4/21 cell line and promote *ALKBH5* expression. Therefore, the 3D4/21 cell line was used for analysis thereafter.

### 2.3. Disruption of ALKBH5 Expression Increases PEDV Infection Capacity

To clarify the function of ALKBH5 for PEDV infection, an RNA interference experiment was performed to disrupt its expression in 3D4/21 cells (Figure 3a). We obtained an interference efficiency of higher than 80% (Figure 3b), and Western blotting confirmed that its protein abundance was also dramatically reduced (Figure 3c). These results indicate that the *ALKBH5* gene interference was successfully conducted and could be used for the next assay. The expression levels of multiple apoptosis-related (*Caspase 3*, *Caspase 8*, and *Caspase 9*) and inflammation-related (*IL-6*, *IL-1β*, and *IFN-β*) genes were significantly down-regulated, and the *Bcl-2/Bax* ratio was significantly increased (Figure 3d,e), which is in contrast to the trend observed during PEDV infection (Figure 2c,d). After determining that the PEDV copy number significantly decreased after infection for 36 h (Figure 3f), we further examined how PEDV infection could be influenced after the disruption of *ALKBH5* expression. Interestingly, our results clearly demonstrate that the disruption of *ALKBH5* expression results in an increase in the infection’s capacity for PEDV (Figure 3g), suggesting that ALKBH5 may play an important role in restraining PEDV infection.

### 2.4. Transcriptome Analysis Identified GAS6 as a Putative Target Gene of ALKBH5

After revealing that the knockdown of *ALKBH5* affected the infection capacity for PEDV, the exact molecular mechanism remained to be explored. We applied RNA-seq to determine the genome-wide gene expression alterations after *ALKBH5* knockdown, which identified a total of 175 differentially expressed genes (DEGs), including 97 down-regulated and 78 up-regulated genes (Figure 4a,b). Ten DEGs (*CCL2*, *CHAC1*, *MMP7*, *TGFBI*, *DPP4*, *SFTA2*, *ID3*, *SCG2*, *TMEM160*, and *GAS6*) were randomly selected, with their expression alterations confirmed by qPCR (Figure 4c). GO enrichment analysis showed that these DEGs are associated with the immune system processes, virion parts, enzyme regulator activity, fusion of virus membrane with the host plasma membrane, viral genome replication, and receptor-mediated virion attachment to host cells (Figure 4d). Notably, these pathways, which are associated with viral infection, involve the *GAS6* gene. Growth arrest-specific 6 (Gas6) is a member of the vitamin K-dependent protein family, which is closely associated with viral infection [21]. Together, transcriptome analysis identified hundreds of genes that are probably under the regulation of ALKBH5, including *GAS6*, which is a viral infection-related gene that we selected for further analysis.

### 2.5. ALKBH5 Depends on YTHDF2 to Affect the Expression of Target Gene GAS6

We examined the expression of *GAS6* after PEDV infection in 3D4/21 cells and found that, similar to *ALKBH5* (Figure 2e), the expression of *GAS6* is also dramatically up-regulated (Figure 5a). *GAS6* expression was remarkably decreased after the disruption of *ALKBH5* (Figure 5b), which indicated a positive correlation with its expression. Using SRAMP [22] and RMBasev2.0 [23], we inferred that there were two m^6^A modification sites (c.1444 bp and c.1469 bp) in the coding region of *GAS6* (Figure 5c−e). MeRIP-qPCR further demonstrated that the m^6^A level of *GAS6* was significantly increased after the disruption of *ALKBH5* (Figure 5f,g), which was expected because ALKBH5 is a demethylase. These results suggest that ALKBH5 reduced the m^6^A level of *GAS6*, which probably, in turn, increased its expression.

To learn more about the regulatory mechanism of ALKBH5 on *GAS6*, we further examined N^6^-methyladenosine RNA-binding protein 2 (YTHDF2), which is known to recognize and degrade m^6^A-modified mRNAs [14]. To determine whether the GAS6 transcript might be degraded by YTHDF2, we knocked down the expression of YTHDF2 and examined how the *GAS6* expression was changed (Figure 5h,i). As expected, the disruption of *YTHDF2* expression significantly increased the *GAS6* expression levels (Figure 5j), indicating that YTHDF2, indeed, regulated *GAS6*. Interestingly, the disruption of YTHDF2 also significantly reduced the PEDV infection capacity (Figure 5k). The measurement of *GAS6* mRNA decay after blocking new RNA synthesis with actinomycin D showed that silencing *ALKBH5* significantly improved GAS6 mRNA stability (Figure 5l). Taken together, these results suggest that ALKBH5 restrains PEDV infection, at least partly, via the promotion of *GAS6* expression through an m^6^A-YTHDF2-mediated mechanism.

## 3. Discussion

The PEDV epidemic has caused serious economic losses to the pig industry worldwide. The most prominent clinical symptom of PEDV infection in piglets is watery diarrhea. Thus, it is generally believed that its pathogenic pathway is limited to the gastrointestinal tract. However, based on the previously reported airborne properties of PEDV [4,24], we speculate that the respiratory tract may also be involved in its infection. In this study, we found that, although infected piglets did not show symptoms of primary respiratory disease, PEDV could still be detected in their lung tissue. Through the cell infection experiment, we found that the classic PEDV strain CV777 could infect porcine alveolar macrophages and induce an obvious cytopathic effect. Park et al. reported that cell-adapted PEDV isolates KPEDV-9 and SM98LVec could also infect porcine alveolar macrophages [25], which are the key immune cells in the alveoli [26]. Therefore, it is speculated that PEDV may escape into the alveolar macrophages when vaccinated or medically treated; thus, it has the potential to cause a persistent infection in the host. It has previously been reported that PEDV infection caused by inoculation through the upper respiratory tract (nasal cavity) is due to PEDV-carrying dendritic cells (DCs), thus allowing the virus to transfer to CD3^+^ T cells through the virological synapse, which, in turn, reach the intestines through blood circulation, causing infection [4]. Our results further confirm that infection with PEDV could also occur in the lower respiratory tract, including the lung tissue and alveolar macrophages.

m^6^A modification has been shown to regulate viral replication. Studies have shown that methyltransferase METTL3/14 and demethylase FTO are involved in the regulation of PEDV replication in LLC-PK1 cells [20]. However, the mechanism of m^6^A modification in other tissues or cell types (outside the gastrointestinal tract) of PEDV infection is still unclear. Here, we detected the expression of different m^6^A modifiers in the lung tissue of PEDV-infected piglets and found that only PEDV infection significantly induced the expression of *ALKBH5*. Recent studies have shown that ALKBH5 can control the immune response. ALKBH5 can regulate the production of type I interferons, which are triggered by dsDNA or human cytomegalovirus [27]. ALKBH5 controls the pathological effects of CD4^+^ T cells during autoimmunity [28]. RNA helicase DDX46 inhibits innate immunity by recruiting ALKBH5 to erase the m^6^A modification of antiviral transcripts, thus strengthening their retention in the nucleus [29]. Notably, we measured the expression of *ALKBH5* genes in different tissues of 7-day-old normal piglets and found that the *ALKBH5* gene was highly expressed in the lungs and intestines. Therefore, we speculate that ALKBH5 may play a special role in the immunity regulation of piglets. The ALKBH5 expression in 3D4/21 cells also increases after PEDV infection. After interfering with ALKBH5, many apoptosis- and inflammation-related genes were significantly down-regulated, and the PEDV replication level was significantly increased. For the relationship between ALKBH5 and viral replication, it was found that HIV-1 replication significantly increased after silencing ALKBH5 [16], whereas ZIKV envelope protein expression significantly decreased [19]. Overall, our study suggests that the m^6^A demethylase ALKBH5 plays an important role in restraining PEDV infection.

To deeply interrogate how *ALKBH5* mediates m^6^A modification to regulate PEDV resistance, we applied RNA-seq to determine *GAS6* as one of the putative target genes regulated by ALKBH5. Growth arrest-specific 6 (Gas6) is a vitamin K-dependent secreted protein that plays a key role in reducing TLR signaling, thus promoting the phagocytic internalization of apoptotic bodies and production of inflammatory cytokines [30,31,32]. Previous studies have also reported that the Gas6/TAM receptor signaling pathway is associated with the development of a variety of diseases, such as autoimmune diseases, tumors, and fibrosis [33,34,35]. In addition, it has been shown that Gas6 is a key regulator of antimicrobial immunity after primary respiratory syncytial virus infection [36]. In vitro studies have shown that Gas6 binds to adenoviral capsids in a serotype-dependent manner [37]. The soluble protein Gas6 binds to phosphatidylserine on the virion surface and bridges the virus to the cell surface [38]. Circulating levels of Gas6 increase in the serum of ZIKV-infected patients [39]. In this study, we used qPCR, Western blot, and MeRIP-qPCR validation to determine that ALKBH5 had a significant effect on both *GAS6* expression and m^6^A modification levels. All these studies suggest that *GAS6* may be a key target gene of ALKBH5, which may be important for mediating the function of ALKBH5 in restraining the PEDV infection in porcine alveolar macrophages.

The biological function of the m^6^A modifications is highly dependent on m^6^A “readers”, which selectively bind methylated RNAs and determine the fate of the transcripts [40]. YTHDF1 reader protein recognition can promote the transcription and translation of m^6^A-modified mRNA, whereas YTHDF2 recognition can affect the mRNA stability of the target genes and reduce the translation efficiency [41]. We found that, after *ALKBH5* interference, the modification level of target gene *GAS6* m^6^A significantly increased, and it significantly decreased the transcript level of the target gene *GAS6*, whereas the deletion of *YTHDF2* significantly increased the expression level of *GAS6* in 3D4/21 cells and significantly reduced the infection of PEDV. Thus, it is speculated that YTHDF2 may act as a key reader protein for the ALKBH5-mediated regulation of PEDV replication via m^6^A modification. We measured the decay of *GAS6* mRNA, after blocking novel RNA synthesis using actinomycin D, and found that silencing *ALKBH5* significantly improved *GAS6* mRNA stability. These results indicate that YTHDF2 plays an important role in the ALKBH5-mediated regulation of *GAS6* expression.

In summary, our results suggest that the m^6^A demethylase ALKBH5 restrains PEDV infection, at least partly, by influencing *GAS6* expression via an m^6^A-YTHDF2-orchestrated manner (Figure 6). These results will help us to understand the role of m^6^A during PEDV infection and provide new insights for dissecting the molecular regulatory mechanism of viral diarrhea in piglets. We also expect that this study will provide valuable candidate genes for epidemic diarrhea resistance in weaned piglets, which would warrant further investigation and provide a theoretical basis for formulating molecular breeding strategies against viral piglet diarrhea in the future.

## 4. Materials and Methods

### 4.1. Experimental Animals and Sample Collection

The piglets (Duroc × Landrace × Yorkshire) used in the experiment were obtained from (Changzhou Fenghua Animal Husbandry Co., Ltd., Changzhou, China), and the same feeding procedures were adopted for five 7-day-old piglets that had the typical symptoms of PEDV infection (depression, emaciation, and watery diarrhea) and five normal 7-day-old piglets, all of which were allowed free access to feed and water; they were humanely slaughtered by the intravenous injection of pentobarbital sodium. Tissue samples of the heart, liver, spleen, lungs, kidney, stomach, muscle, thymus, lymph, duodenum, jejunum, and ileum were collected, immediately stored in liquid nitrogen at the scene, and brought back to the laboratory for storage at −80 °C until use.

### 4.2. Proliferation and Isolation of PEDV

Vero cells (ATCC, CCL-81) were inoculated on 6-well plates at a density of 2.0 × 10^5^ cells per well and cultured using a complete medium composed of DMEM (Gibco, Grand Island, NY, USA), 10% fetal bovine serum (FBS) (Gibco, Grand Island, NY, USA), antibiotics (penicillin (100 U/mL), and streptomycin (0.1 mg/mL) (Beijing Solarbio Science & Technology Co., Ltd., Beijing, China)). When the cell density reached 90%, the PEDV CV777 strain (kindly provided by China Agricultural University) was inoculated, and the virus liquid was collected when 80% of Vero cells exhibited cytopathic effects under microscope; then, they were stored at −80 °C.

### 4.3. Virus Titer Determination

The virus infection ability (virus titer) was determined by the median cell culture infectious dose (TCID_50_) method. Vero cells were plated into 96-well plates; when the cell density reached approximately 60%, PEDV was inoculated at dilutions of 6 gradients per well; each treatment was repeated 8 times, and 8 blank controls were set. Cytopathic effects were observed and recorded daily.

### 4.4. PEDV Infection of 3D4/21 Cells

Porcine alveolar macrophages (3D4/21) (ATCC, CRL-2843) were seeded in 6-well cell culture plates at 2.0 × 10^5^ cells/well and cultured with complete culture medium composed of RPMI-1640 culture medium (Gibco, Grand Island, NY, USA), 10% fetal bovine serum (FBS) (Gibco, Grand Island, NY, USA), antibiotics (penicillin (100 U/mL), and streptomycin (0.1 mg/mL)) (Beijing Solarbio Science & Technology Co., Ltd., Beijing, China). When the cell coverage rate reached approximately 80%, 3D4/21 cells were infected with the classic PEDV strain CV777 at 500 µL per well of virus solution (MOI = 0.1). The virus inoculum was discarded after adsorption at 37 °C for 2 h, and RPMI-1640 complete culture medium was added to continue the culture. At 12, 24, 36, and 48 h of PEDV infection, the cytopathic conditions were observed under a microscope, and the cells were collected.

### 4.5. Construction of 3D4/21 Cell Line with ALKBH5 Gene Interference

*ALKBH5* gene interference lentiviral (LV3-ALKBH5) and negative control (LV3-NC) vectors were synthesized by (GenePharma, Suzhou, China). The 3D4/21 cells were inoculated into 12-well plate and cultured in RPMI-1640 culture medium containing 10% FBS in a constant temperature incubator at 5% CO_2_ and 37 °C. When the cells reached approximately 80% confluency, lentivirus LV3-ALKBH5 and empty vector LV3-NC virus were infected, and a blank control group (blank) was established. There were 3 treatment groups, with 4 repeats in each group. After lentiviral transfection of 3D4/21 cells, fluorescence expression was observed after 48 h of culture in 5% CO_2_ at 37 °C, followed by positive cell selection with 10 μg·mL^−1^ puromycin every 24 h, until all the blank cell groups died and drug addition was stopped.

### 4.6. RNA Extraction and cDNA Synthesis

Total RNA from 3D4/21 cells in various tissues of ternary piglets and at different PEDV infection time points was extracted using TRIzol reagent (Invitrogen, Carlsbad, CA, USA). The degree of RNA integrity was detected by 1% formaldehyde denaturing agarose gel electrophoresis; after the concentration and purity were determined, using a ND-1000 nucleic acid/protein concentration tester, the samples were stored in an ultra-low-temperature freezer at −70 °C until use.

The reverse transcription experiment was performed using a reverse transcription kit (Vazyme Biotech Co., Ltd., Nanjing, China). The extracted total RNA was used as a template to synthesize cDNA: 10 μL of the reaction system contained 2 μL of 5 × qRT SuperMix II, 500 ng of total RNA, and RNase-free ddH_2_O made up to 10 μL. The reaction program was 10 min at 25 °C, 30 min at 50 °C, 5 min at 85 °C, and then stored at 4 °C.

### 4.7. RT-PCR

To further confirm whether piglets with typical symptoms of PEDV infection and cells with cytopathic effects were simply infected with PEDV or had a mixed infection with other common porcine enteric diseases, such as porcine deltacoronavirus (PDCoV), porcine rotavirus (PoRV), or transmissible gastroenteritis virus (TGEV), we used the cDNA obtained by reverse transcription as a template to design specific primers (Appendix A) for RT-PCR amplification, with reference to the reported PEDV-M, PDCoV-N, PoRV-VP6, and TGEV-S1 gene sequences in the GenBank database; the products were detected by agarose gel electrophoresis.

### 4.8. qPCR

According to the gene sequences published in the GenBank database, qPCR primers were designed using the Primer Premier 5.0 software. *GAPDH* was used as a housekeeping gene for input normalization. All primer synthesis was conducted by (Sangon Biotech, Shanghai, China), and the corresponding sequences are shown in Appendix A. qPCR analysis was performed using a real-time fluorescence quantification kit (Vazyme Biotech Co., Ltd., Nanjing, China). All qPCR reactions were conducted in a 20 µL volume composed of 2 µL of cDNA, 0.4 µL of each primer (10 µmol/L), 10 µL of 2 × AceQ Universal SYBR qPCR Master Mix, and 7.2 µL of ddH_2_O. Thermocycler settings were as follows: 95 °C for 5 min, 40 cycles of 95 °C for 5 s, and 60 °C for 30 s. Melting curves were then used to confirm amplified product specificity. Three independent experimental replicates were conducted for all analyses.

### 4.9. Detection of Copy Number of PEDV-M Gene

According to the PEDV genome information, the fluorescent quantitative primers of M gene were designed (Appendix A), and the number of PEDV M gene cycles was detected by fluorescence quantitative analysis. The equation of the standard PEDV CV777 curve was established previously: y = −3.3354 lg(x) + 37.832, R2 = 0.9994, and the PEDV copy number was calculated [42,43].

### 4.10. Western Blotting

The collected cells were washed with precooled PBS at 4 °C, and a mixture of RIPA lysis buffer and protease inhibitor cocktail (Labgic Technology Co., Ltd., Hefei, China) was added, lysed on ice for 20 min, and centrifuged at 14,000 rpm for 20 min at 4 °C; then, the supernatant was taken to determine the protein concentration with a BCA protein assay kit (Beyotime, Shanghai, China). Denaturing was performed with 5 × SDS–PAGE loading buffer at 98 °C for 10 min. After sodium dodecyl sulfate–polyacrylamide gel electrophoresis (SDS–PAGE), the sample was transferred to a polyvinylidene fluoride membrane (PVDF) (Millipore, Shanghai, China). Blocking was performed with 5% skimmed milk for 2 h at room temperature, and the blocked membranes were incubated with primary antibodies (ALKBH5 antibody, abcam, Shanghai, China; YTHDF2 antibody, abcam, Shanghai, China; HSP90 antibody, abcam, Shanghai, China) overnight at 4 °C. After the membranes were washed with TBST, they were incubated with a secondary antibody (Proteintech Group, Wuhan, China) for 1 h at room temperature, washed with TBST, and the proteins were visualized by ECL.

### 4.11. RNA Sequencing

Total RNA was extracted from experimental samples using the TRIzol reagent; the RNA purity and concentration were initially measured using a NanoDrop 2000 spectrophotometer (Thermo Scientific, CA, USA), and the RNA integrity was accurately quantified using an Agilent 2100 bioanalyzer (Agilent Technologies, Santa Clara, CA, USA). Samples with an RNA integrity number (RIN) ≥ 7 were subjected to subsequent analysis. mRNA was enriched, and double-stranded cDNA was synthesized; the end was repaired, the sequencing linker PolyA was added, PCR amplification of the product was performed, and the library was sequenced with an Illumina HiSeqTM 2500 system (Illumina, San Diego, CA, USA). The raw data on the machine were subjected to quality control and data filtering, and the processed reads were aligned to the pig reference genome (release Sscrofa11.1) using HISAT2 [44]. Differentially expressed genes were identified by DESeq2 [45], with cut-offs of FDR < 0.05 and FoldChange ≥ 2. The differentially expressed gene results were validated by qPCR (Appendix A).

### 4.12. Methylated RNA Immunoprecipitation Coupled with qPCR (MeRIP-qPCR)

After the cells were collected, they were rinsed with precooled PBS 3 times. The total RNA of the cells was extracted with TRIzol reagent, following the manufacturer’s instructions. The integrity of RNA was detected by formaldehyde denaturing agarose gel electrophoresis. RNA was fragmented using a non-contact, fully-automatic ultrasonic crusher (Covaris, Woburn, MA, USA). Small fractions (10%) of the fragmented RNA fragments were collected as input samples (input). Fragmented RNA was incubated and immunoprecipitated with m^6^A antibody (Cell Signaling Technology, Danvers, MA, USA) for 4 h at 4 °C with rotation. The m^6^A antibody RNA mixture was incubated with protein A/G magnetic beads for 2 h at 4 °C with rotation. RNA was extracted using TRIzol reagent, and the enrichment of m^6^A was detected with quantitative reverse transcription polymerase chain reaction (qPCR). The primers used for MeRIP-qPCR are listed in Appendix A.

### 4.13. mRNA Stability Analysis

To determine mRNA stability, cells were treated with actinomycin D (MedChemExpress, Shanghai, China) at a final concentration of 10 μg/mL for 0, 3, 6, and 9 h [46]. Cells were collected, and RNA was extracted; the mRNA transcription levels of the target genes were detected by qPCR.

### 4.14. Statistical Analyses

The results of relative quantification were analyzed and processed using the 2^−ΔΔCt^ method [47], and the expression levels were normalized to appropriate internal control genes. SPSS 25.0 software (SPSS, Inc., Chicago, IL, USA) was used to compare data via ANOVA with LSD tests, and test data are expressed as the mean ± standard deviation (mean ± SD), with multiple replicates for each independent experiment and treatment. A *p* value of < 0.05 was considered statistically significant. * *p* < 0.05, ** *p* < 0.01, *** *p* < 0.001.

## Figures and Tables

**Figure 1 ijms-23-06191-f001:**
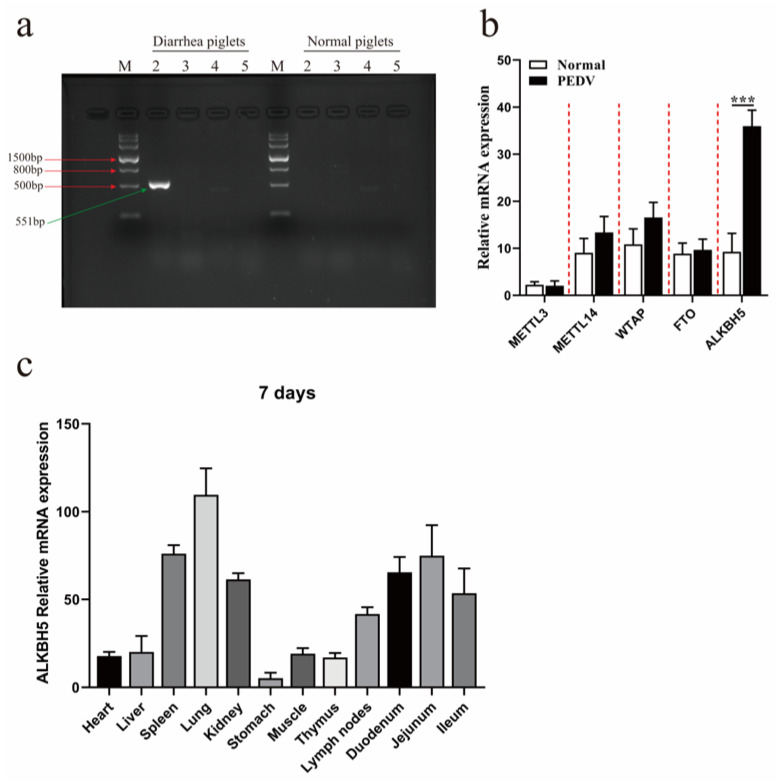
Increased expression of *ALKBH5* in the lung tissues of piglets after PEDV infection. (**a**) Gel picture for PCR results against distinct marker genes against different viruses in the lung tissues of individual piglets with diarrhea, as well as normal piglets. M: Marker III molecular weight standard. Lanes 2–5 represent PEDV-M, PDCoV-N, PoRV-VP6, and TGEV-S1, respectively. (**b**) Expression level of m^6^A-related modifying enzyme genes in lung tissues of normal and PEDV-infected piglets. Normal: normal piglets; PEDV: PEDV-infected piglets. (**c**) Expression profile of *ALKBH5* in twelve different tissues of normal 7-day-old piglets. At least three biological replicates are analyzed for each group. Data are presented as the mean ± SD; *** *p* < 0.001.

**Figure 2 ijms-23-06191-f002:**
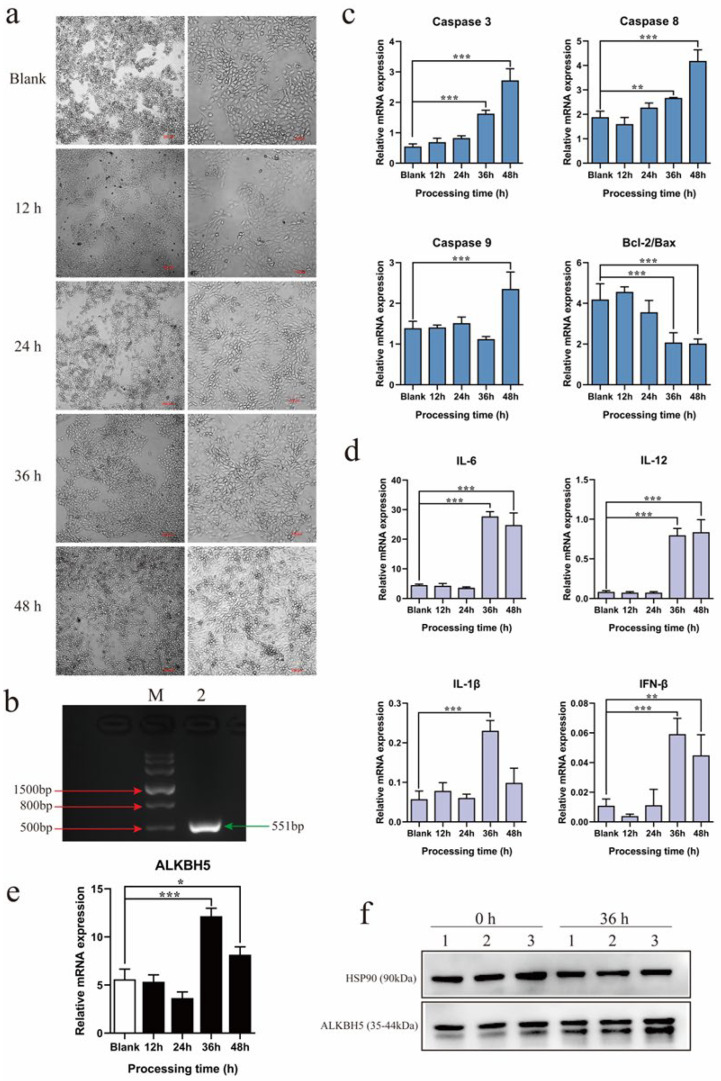
Effects of PEDV infection on 3D4/21 cells and ALKBH5 expression. (**a**) Phenotype of 3D4/21 cells during PEDV infection by optical microscopy. Blank: 3D4/21 cells without PEDV infection. (**b**) Agarose gel assay of PCR results against hallmark genes of viruses in PEDV-infected 3D4/21 cells. M: Marker III molecular weight standard. Lane 2 represents PEDV-M. (**c**) Effect of PEDV infection on the expression of apoptosis-related genes in 3D4/21 cells. Blank: 3D4/21 cells without PEDV infection. (**d**) Effects of PEDV infection on the expression of inflammation-related genes in 3D4/21 cells. Blank: 3D4/21 cells without PEDV infection. (**e**) ALKBH5 expression in 3D4/21 cells infected with PEDV. Blank: 3D4/21 cells without PEDV infection. (**f**) Western blotting for ALKBH5 protein abundance in 3D4/21 cells infected with PEDV for 36 h. At least three biological replicates were used for each group. Data are presented as the mean ± SD; * *p* < 0.05, ** *p* < 0.01, *** *p* < 0.001.

**Figure 3 ijms-23-06191-f003:**
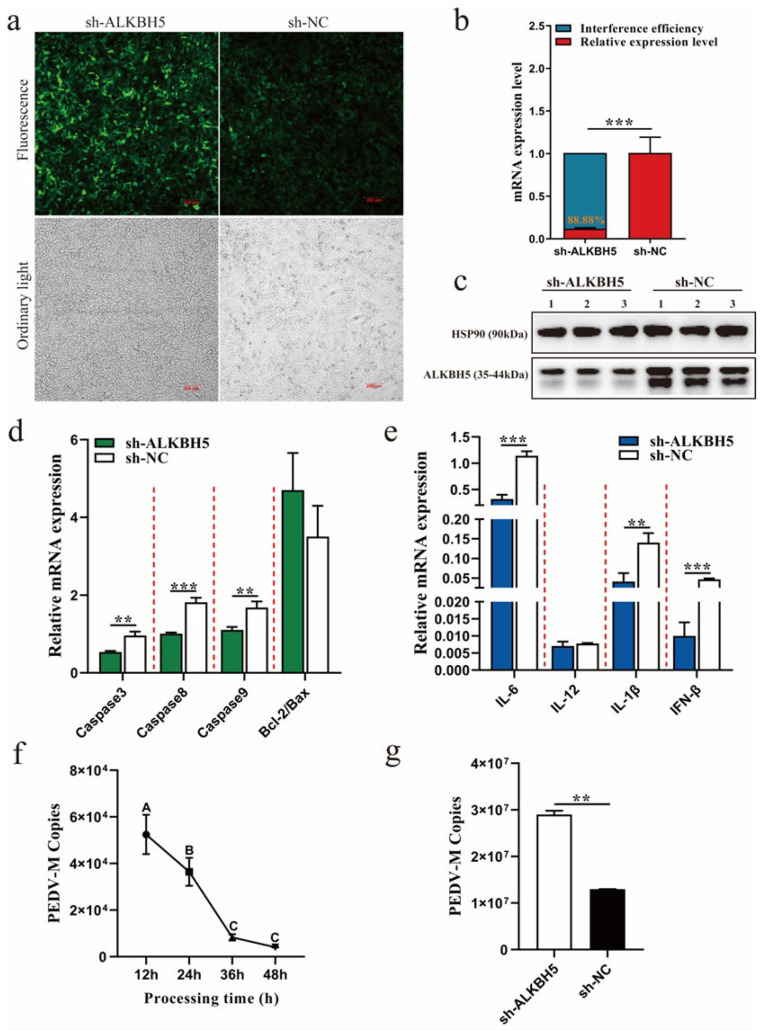
Disruption of ALKBH5 expression by RNAi increases PEDV infection capacity. (**a**) Expression of GFP in lentivirus-transfected 3D4/21 cells screened by puromycin. (**b**) qPCR was performed to detect *ALKBH5* gene interference efficiency. sh-ALKBH5: lentivirus-transfected LV3-ALKBH5 group. sh-NC: lentivirus-transfected LV3-NC group. (**c**) The interference efficiency of ALKBH5 was detected by Western blotting. (**d**) Effect of interfering with the *ALKBH5* gene, regarding the expression of apoptosis-related genes in 3D4/21 cells. (**e**) Interfering with the effect of the *ALKBH5* gene on the expression of inflammation-related genes in 3D4/21 cells. (**f**) Copy number of PEDV-infected 3D4/21 cells at different time periods. (**g**) Effect of *ALKBH5* gene interference on PEDV copy number. At least three biological replicates are used for each group. Data are presented as the mean ± SD; ** *p* < 0.01, *** *p* < 0.001.

**Figure 4 ijms-23-06191-f004:**
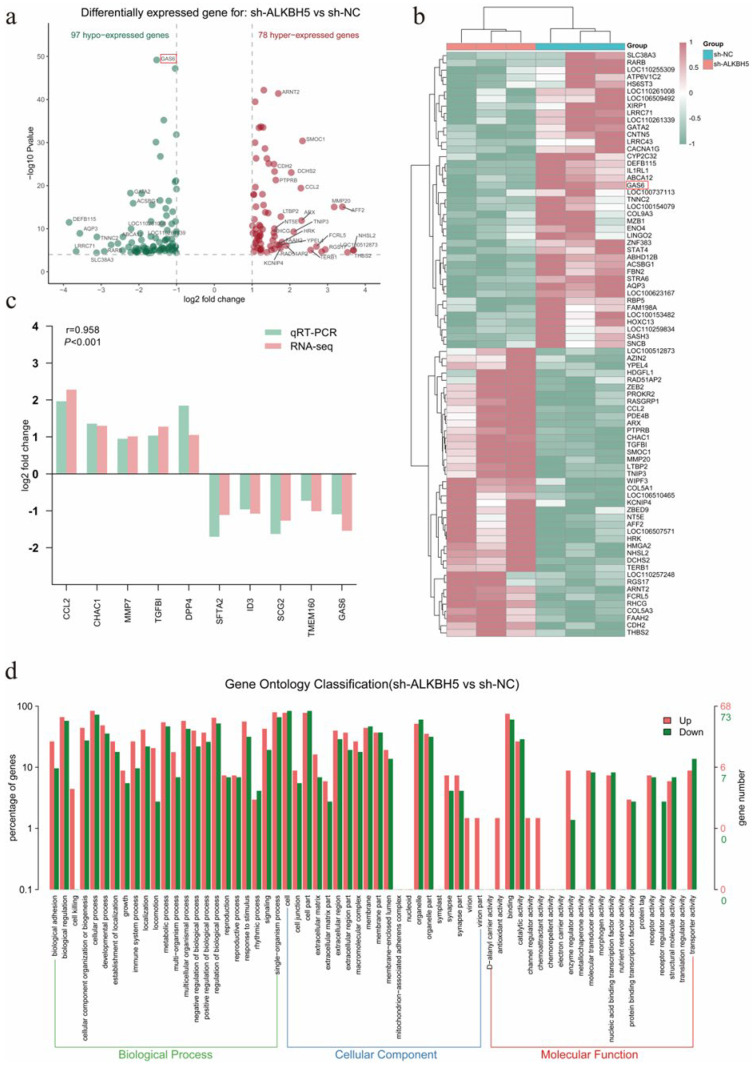
Disruption of *ALKBH5* expression by RNAi causes the altered expression of hundreds of genes in 3D4/21 cells. (**a**,**b**) Volcano plot and heatmap of differentially expressed genes after *ALKBH5* gene interference. (**c**) qPCR validation of differentially expressed genes. (**d**) GO enrichment analysis of differentially expressed genes.

**Figure 5 ijms-23-06191-f005:**
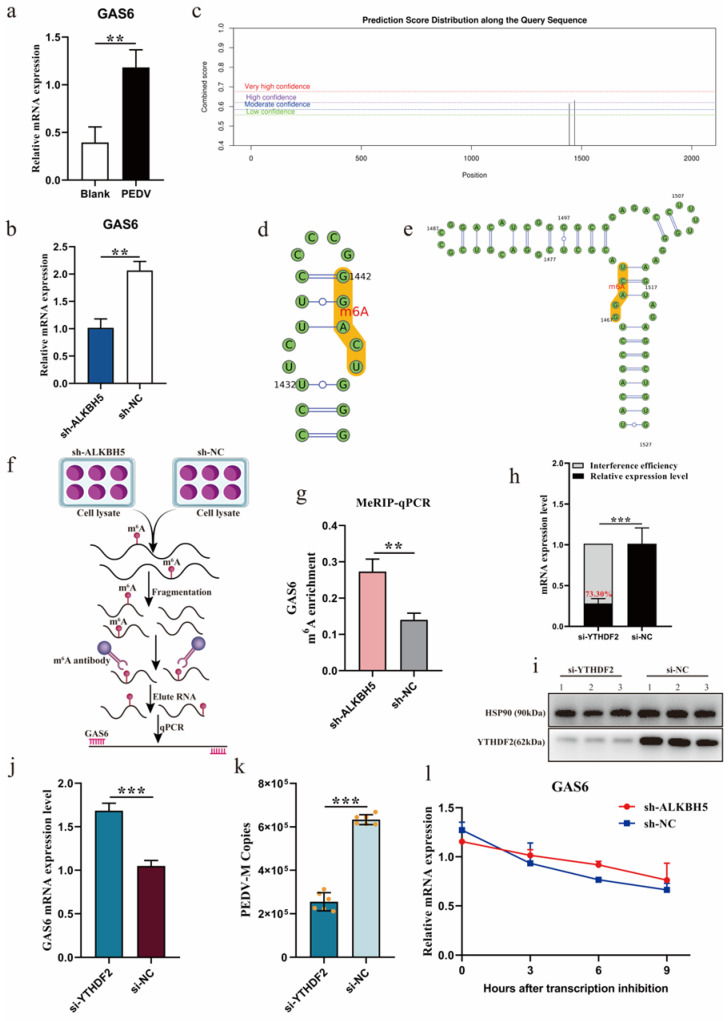
ALKBH5 regulates *GAS6* expression through an m^6^A-YTHDF2-mediated mechanism. (**a**) Detection of *GAS6* expression levels before and after PEDV infection of 3D4/21 cells. Blank: 3D4/21 cells without PEDV infection. (**b**) Detection of target gene *GAS6* expression levels before and after *ALKBH5* interference. (**c**) Prediction of m^6^A modification sites in the coding region of target gene *GAS6.* (**d**,**e**) The secondary structure of the m^6^A modification site of the *GAS6* gene was predicted. (**f**) Flow chart of the methylated RNA immunoprecipitation (MeRIP) technique. (**g**) Detection of m^6^A levels of *GAS6* in the sh-ALKBH5 and sh-NC groups by MeRIP-qPCR. (**h**) qPCR analysis of interference efficiency of *YTHDF2* gene. (**i**) Western blot analysis of *YTHDF2* gene interference efficiency. (**j**) Analysis of *GAS6* expression levels after *YTHDF2* gene interference. (**k**) The effect of *YTHDF2* gene interference on the number of PEDV copies. (**l**) Analysis of mRNA stability of *GAS6* in the sh-ALKBH5 and sh-NC groups treated with actinomycin D for 3, 6, and 9 h. At least three biological replicates were used for each group. Data are presented as the mean ± SD; ** *p* < 0.01, *** *p* < 0.001.

**Figure 6 ijms-23-06191-f006:**
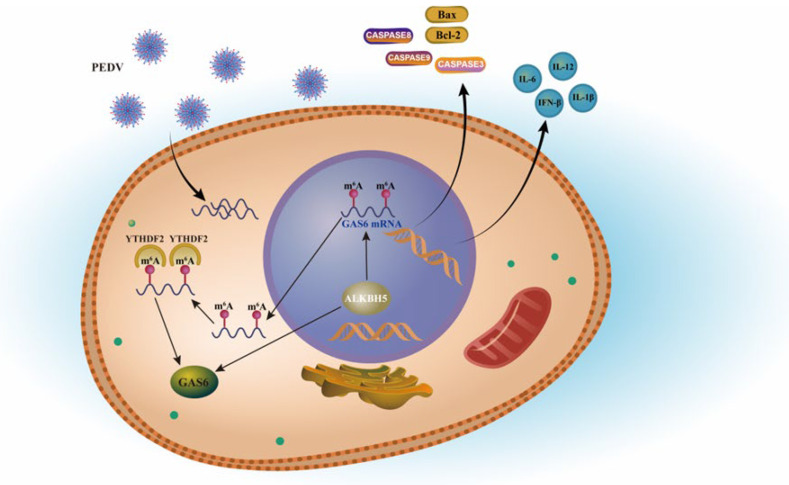
Hypothesis diagram showing that ALKBH5 restrains PEDV infection by influencing the expression of *GAS6*.

## Data Availability

The RNA-seq data generated in this study have been uploaded to the National Center for Biotechnology Information (NCBI), under the BioProject accession number PRJNA822686. (https://dataview.ncbi.nlm.nih.gov/object/PRJNA822686?reviewer=jnepmgte49bf0at445g0g49ckv, accessed on 1 August 2022).

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
