# Peer review of "m6A Demethylase ALKBH5 Restrains PEDV Infection by Regulating GAS6 Expression in Porcine Alveolar Macrophages"

_ijms, 2022, doi:10.3390/ijms23116191_

Round 1

Reviewer 1 Report

Jin et al investigated the molecular mechanisms of PDEV infection on alveolar macrophages, and reported the involvement of the ALKBH5 and its projected target GAS6.

Introduction:

The information conveyed in the introduction is appropriate, however, found the second paragraph confusingly communicated. This could do with a rewrite and further clarification.

Results:

The label ‘Blank’ is used in many of the figures – what does this mean?  If it is a true ‘blank’ then this is not a suitable control to use.  Do you mean 0hr time point (Figure 2 c, d, e) or empty vector (Figure 5a) samples?

Figure 1b: legend reads ‘before and after PEDV infection – this is confusing as the samples are normal and unhealthy, not as the current legend implies, samples taken from the same piglets before and after PEDV infection.

ALKBH5 expression data is provided for numerous healthy pig tissues but not tissues from PEDV infected animals. As the study is investigating the link between PEDV infectivity and ALKBH5 this seems like an obvious omission – it is unclear if PEDV infection influences ALKBH5 expression which is critical to interpreting the provided data.

Figure 2 b shows PCR results of supposed cell samples after exposure to CV777 – why were the other viruses tested for? I can understand this strategy in the piglet sample, where the infectious agent unknown, but seems strange testing this in cells that have only been exposed to one virus type.

Figure 2f, 3c, shows western blots for ALKBH5, with 2 bands present. Which one represents ALKBH5? Please indicate on figures.  If both bands represent different forms of ALKBH5 then please explain in manuscript.

Data shown in figure 3a-e and figure 4 - are these samples infected with PEDV?  This is not explicitly stated, however experimentally it does not make sense to perform these experiments on normal cells in the absence of PEDV infection. You cannot directly extrapolate downstream effects of ALKBH5 knockdown on uninfected cells to infected cells.

At what time period was the data presented in Fig 3g taken?

What are the Gas6 levels in PEDV infected cells with and without ALKBH5 knockdown? This data would help to tie together your conclusion however is noticeably absent.

Author Response

Response to reviewer’s comments:

Introduction:

The information conveyed in the introduction is appropriate, however, found the second paragraph confusingly communicated. This could do with a rewrite and further clarification.

Response:

Thanks for your advice. We have extensively re-wrote that paragraph as you suggested, and we believe it is clearer at current form now. The updated paragraph is as below:

“RNA modifications, which play important roles in the regulation of many biological processes, have received increased attentions in recent years [8]. Among them, N6-methyladenine (m6A) is the most abundant form [9] and it exists not only in the transcripts of eukaryotic and DNA viruses, but also in the genome of RNA virus [10]. m6A modifications are highly reversible, with their patterns established by so-called “writers” including METTL family proteins and Wilms' tumor 1-associating protein (WTAP) [11], and removed by “erasers” including obesity-associated protein (FTO) and alkB homologue 5 (ALKHB5) [12,13]. Furthermore, the YTH domain family (YTHDF) protein serves as the "reader" to selectively bind methylated RNA molecules and mediates downstream process [14,15]. m6A modifications play important roles in viral replication and host immune responses. For example, silencing of ALKHB5 in human T cells increases the replication of human immunodeficiency virus type 1 (HIV-1) [16], and inactivation of METTL3 in A549 cells inhibits influenza A virus (IAV) replication [17]. Other studies demonstrate that the overexpression of YTHDF2 promotes the replication of simian virus 40 (SV40) [18], while m6A modifications inhibit the lytic replication of ZIKV [19]. Recent evidence indicates that m6A modifications modulate the replication of PEDV [20], yet that study is based on results from green monkey Vero kidney cells and porcine LLC-PK1 kidney epithelial cells. However, the mechanisms of m6A modification for PEDV infection in other tissues or cell types outside the gastrointestinal tract remain to be explored.”

Results:

The label ‘Blank’ is used in many of the figures – what does this mean?  If it is a true ‘blank’ then this is not a suitable control to use. Do you mean 0hr time point (Figure 2 c, d, e) or empty vector (Figure 5a) samples?

Response:

Sorry for confusing. The 'Blank' tags used in Figures 2a, c, d, e and Figure 5a indicate the group without PEDV infection. We have explained it in the figure legend now.

Figure 1b: legend reads ‘before and after PEDV infection – this is confusing as the samples are normal and unhealthy, not as the current legend implies, samples taken from the same piglets before and after PEDV infection.

Response:

Thanks for your comments. We have corrected that sentence as "Expression levels of m6A-related modifying enzyme genes in lung tissues of normal piglets and PEDV-infected piglets " (Line 96-98).

ALKBH5 expression data is provided for numerous healthy pig tissues but not tissues from PEDV infected animals. As the study is investigating the link between PEDV infectivity and ALKBH5 this seems like an obvious omission – it is unclear if PEDV infection influences ALKBH5 expression which is critical to interpreting the provided data.

Response:

Thanks for your insightful comments. This study focused on the potential novel transmission routes of PEDV (porcine lung tissue and porcine alveolar macrophages). As you can see, in Figure 1b we indeed compared the expression of different m6A modifiers (ie. METTL3, METTL14, WTAP, FTO and ALKBH5) in the lung tissue of healthy piglets and PEDV-infected piglets, and we found that ALKBH5 is the only one got significantly up-regulated after PEDV infection. Therefore, we selected ALKBH5 as the candidate gene for in-depth study. Based on this result, we then further explored the expression profile of ALKBH5 in different tissues of healthy pigs (Figure 1c).

Figure 2 b shows PCR results of supposed cell samples after exposure to CV777 – why were the other viruses tested for? I can understand this strategy in the piglet sample, where the infectious agent unknown, but seems strange testing this in cells that have only been exposed to one virus type.

Response:

Thanks for your comments. We tested other viruses (PDCoV and TGEV) in order to exclude potential contamination from them, since our neighboring labs are working on these two viruses. Now we realize it probably is not necessary to include such results. Now we have modified Figure 2b to only keep PEDV.

Figure 2f, 3c, shows western blots for ALKBH5, with 2 bands present. Which one represents ALKBH5? Please indicate on figures.  If both bands represent different forms of ALKBH5 then please explain in manuscript.

Response:

In Figure 2f, 3c, both bands are for ALKBH5 and they are caused by protein homodimers. More details can be found from the website of the used antibody: https://www.abcam.cn/alkbh5-antibody-epr18958-ab195377.html#lb.

Data shown in figure 3a-e and figure 4 - are these samples infected with PEDV? This is not explicitly stated, however experimentally it does not make sense to perform these experiments on normal cells in the absence of PEDV infection. You cannot directly extrapolate downstream effects of ALKBH5 knockdown on uninfected cells to infected cells.

Response:

Yes, the samples for Figure 3a-e and Figure 4 are uninfected PEDV. Since at the beginning (Figures 1 and 2) we have already demonstrated that ALKBH5 can affect PEDV infection, in the following part (Figures 3, 4 and 5) we aim to further investigate the regulatory mechanism of ALKBH5 and screen for downstream regulatory genes of ALKBH5 in the host by using transcriptome profiling. For these reasons, we did not use PEDV for infection experiments subsequently.

At what time period was the data presented in Fig 3g taken?

Response:

The data shown in figure 3g were collected 36 h after PEDV infection. We have described this in the manuscript (Line 139-144).

What are the Gas6 levels in PEDV infected cells with and without ALKBH5 knockdown? This data would help to tie together your conclusion however is noticeably absent.

Response:

Thanks for your comments. Figure 5a presents the expression level of Gas6 in PEDV-infected cells without knockdown of ALKBH5. The focus of this study was to investigate the mechanism by which ALKBH5 regulates Gas6 expression which in turn affects PEDV infection, so the expression level of Gas6 in PEDV-infected cells that were knocked down by ALKBH5 was not examined.

Reviewer 2 Report

In this manuscript, Jin et al. investigated the N6-methyladenosine (m6A) in Porcine epidemic diarrhea virus (PEDV) infection. They discovered the induction of ALKBH5 in macrophase cell lines after infection, which is the primary m6A demethylase. Transcriptome analyses revealed several potential targets of ALKBH5. The approaches, including qRT-PCR, RNA-seq, RNAi, and western blotting, are solid and comprehensive. This study provides insights into RNA methylation of PEDV, and will inform disease control strategies.

Please find the specific comments below.

There are numerous grammatical errors in this manuscript, and I recommend the authors seek editing help from someone (or an editing service) with full professional proficiency in English to proofread the manuscript. For example, in the abstract, Line 12, “suggest” should be “suggests”.

Please summarize the world-wide or regional economic loss caused by PEDV in the introduction.

Line 83, I was wondering if any histopathology work has been done contrasting the PEDV infection and non-infected lungs?

Line 92, after the piglets were recovered from PEDV infection, is the ALKBH5 expression level in lung back to normal?

Author Response

Response to reviewer’s comments:

There are numerous grammatical errors in this manuscript, and I recommend the authors seek editing help from someone (or an editing service) with full professional proficiency in English to proofread the manuscript. For example, in the abstract, Line 12, “suggest” should be “suggests”.

Response:

Thanks for your comments. We have carefully polished the manuscript by ourselves. We have also sought English editing services to proofread the manuscript.

Please summarize the world-wide or regional economic loss caused by PEDV in the introduction.

Response:

Thanks for your advice. We have summarized this in the introduction section of the manuscript (Line 31-33). Reference regarding the economic loss caused by PEDV is also cited. The added sentence is as below:

“In Asia and Europe, as the main pork production regions in the world, the large-scale outbreak and spread of PEDV have caused huge economic losses to the world pork industry [2]”

Line 83, I was wondering if any histopathology work has been done contrasting the PEDV infection and non-infected lungs?

Response:

Thanks for your comments. We did not perform a histopathological contrast between PEDV-infected and uninfected lungs. This part of the study focused on determining whether PEDV infects lung tissue. In veterinary clinic, RT-PCR as a standard detection technique for the detection of PEDV infection, can quickly and accurately determine the presence or absence of PEDV infection. Your comments are very enlightening for us to carry out relevant researches in the future.

Line 92, after the piglets were recovered from PEDV infection, is the ALKBH5 expression level in lung back to normal?

Response:

Your question is quite interesting. Previously, we detected the expression of ALKBH5 in the lung tissues of 5 piglets with typical symptoms of PEDV infection and 5 normal piglets (Figure 1b), and compared the expression of ALKBH5 in 12 different tissues of normal piglets (Figure 1c). So far, we have not perform following-up experiments use individuals who recovered healthy piglets after PEDV infection for the study. But based on our results from 3D4 cells (Figure 2e), we think it is reasonable to expect the expression level of ALKBH5 would reduce to normal after recovered from PEDV infection.
